# Cytotoxic Effects of Major and Emerging Mycotoxins on HepaRG Cells and Transcriptomic Response after Exposure of Spheroids to Enniatins B and B1

**DOI:** 10.3390/toxins16010054

**Published:** 2024-01-18

**Authors:** France Coulet, Monika Coton, Cristian Iperi, Marine Belinger Podevin, Emmanuel Coton, Nolwenn Hymery

**Affiliations:** 1Univ Brest, INRAE, Laboratoire Universitaire de Biodiversité et Ecologie Microbienne, F-29280 Plouzané, France; france.coulet@univ-brest.fr (F.C.); monika.coton@univ-brest.fr (M.C.); marine.podevin@hotmail.com (M.B.P.); emmanuel.coton@univ-brest.fr (E.C.); 2Autoimmunité et Immunothérapies UMR 51227, Inserm, University Brest, Lymphocytes B, F-29200 Brest, France; cristian.iperi@univ-brest.fr

**Keywords:** enniatins, beauvericin, in vitro models, HepaRG, spheroids, RNA-seq analysis, cytotoxicity

## Abstract

Mycotoxins, produced by fungi, frequently occur at different stages in the food supply chain between pre- and postharvest. Globally produced cereal crops are known to be highly susceptible to contamination, thus constituting a major public health concern. Among the encountered mycotoxigenic fungi in cereals, *Fusarium* spp. are the most frequent and produce both regulated (i.e., T-2 toxin, deoxynivalenol -DON-, zearalenone -ZEA-) and emerging (i.e., enniatins -ENNs-, beauvericin -BEA-) mycotoxins. In this study, we investigated the in vitro cytotoxic effects of regulated and emerging fusariotoxins on HepaRG cells in 2D and 3D models using undifferentiated and differentiated cells. We also studied the impact of ENN B1 and ENN B exposure on gene expression of HepaRG spheroids. Gene expression profiling pinpointed the differentially expressed genes (DEGs) and overall similar pathways were involved in responses to mycotoxin exposure. Complement cascades, metabolism, steroid hormones, bile secretion, and cholesterol pathways were all negatively impacted by both ENNs. For cholesterol biosynthesis, 23/27 genes were significantly down-regulated and could be correlated to a 30% reduction in cholesterol levels. Our results show the impact of ENNs on the cholesterol biosynthesis pathway for the first time. This finding suggests a potential negative effect on human health due to the essential role this pathway plays.

## 1. Introduction

Cereals and their derivatives represent the most important food and feed resources worldwide [1]. In 2021, nearly 8.43 gigatons of cereals were produced globally, among which 741.9 megatons were produced in Europe, positioning the continent in third place after Africa and Asia. France was Europe’s leading producer, with around 235.9 megatons [2].

During crop production and throughout the food supply chain, cereals and cereal-based products can be contaminated by microorganisms, especially fungi that are well adapted to these environments. Many species are xerophilic and can develop at relatively low water activities, but they also colonize these crops by releasing enzymes that hydrolyze various substrates to provide nutrients for growth and also for their metabolism. Among these fungal species, some produce mycotoxins. Mycotoxins, which are defined as secondary (or specialized) metabolites associated with toxic chemical properties, can be produced by various species such as *Aspergillus* spp., *Alternaria* spp., *Fusarium* spp. or *Penicillium* spp. [3,4,5,6]. Environmental conditions such as temperature, humidity and storage conditions all influence fungal growth and potentially mycotoxin production. Mycotoxins are known to be resistant to thermal and chemical treatments and industrial processes [7,8,9]. It is estimated that 25% of world food crops and 20% of crops in the European Union are contaminated by mycotoxins [10]. More than 1000 mycotoxins have been isolated and characterized so far, but only about thirty have been recognized as a threat to humans or animals [11]. A wide range of toxic effects have been reported as hepatotoxic (aflatoxin B1), nephrotoxic (ochratoxin A and citrinin), carcinogenic (aflatoxin B1, ochratoxin A, fumonisin B1), dermonecrotic (trichothecenes), neurotoxic (fumonisin B1 and T-2 toxin), immunosuppressive (aflatoxin B1, ochratoxin A and T-2 toxin), and endocrine disruptors (zearalenone) [12,13,14]. As the main mode of mycotoxin exposure corresponds to the ingestion of contaminated foods, the first organs that are most likely to be affected are the intestines, as the first barrier, and the liver, as the detoxification organ. In this context, some mycotoxins are regulated in some countries, and in Europe, maximum levels have been set for several mycotoxins (aflatoxins -B1, M1, or the sum of B1, B2, G1, and G2-; ochratoxin A; patulin, deoxynivalenol (DON); zearalenone (ZEA); fumonisins or the sum of B1 and B2; citrinine; ergot; and alkaloids T-2 and HT-2) in some foods and feed (Commission Regulation (EU) 2023/915). Members of the *Fusarium* genus are among the most important phytopathogenic fungi present on all continents and among the most important mycotoxin producers, producing six regulated mycotoxins [15,16,17]. Indeed, according to the 2022 DMS report, more than 96% of tested food and feed samples were contaminated by *Fusarium* mycotoxins (including major and regulated mycotoxins) [18]. A meta-analysis of 3291 samples from more than 71 countries showed that the prevalence of fusariotoxins was 61% for DON, 57% for ZEA, 39% for 3-acetyl deoxynivalenol (3-DON), 56% for ENN B1, 52% for ENN B, 35% for ENN A1, 28% for ENN A, 21% for 15-O-acetyl-4-deoxynivalenol (15-ADON), 11% for HT-2, and 9% for T-2 toxins. Moreover, the average concentrations found were high, with 548 µM for DON, 77 µM for ZEA, 98 µM for 3-DON, 192 µM for 15-ADON, 27 µM for ENN B1, 65 µM for ENN B, 14 µM for ENN A1, 4 µM for ENN A, 81 µM for HT-2, and 37 µM for T-2 toxin [18].

Multiple studies have shown the toxic effects of T-2/HT-2 toxins, DON, and ZEA on the intestines and liver. The main effects were high hepatocyte mortality (IC_50_ for DON of 2.83 after 48 h) and a significant increase in oxidative stress (after 5 min at 15 µM) or endocrine hormones in the presence of ZEA [12,19]. On Caco-2 intestinal cells, the IC_50_ values obtained for T-2 toxin and HT-2 toxins after 48 h of exposure were 61.9 and 68.2 nM, respectively. Studies [20,21] reported an IC_50_ of 0.19 µM for T-2 toxin and 7.35 µM for DON after 48 h exposure on HepaRG. Juan-García et al. (2019) reported IC_50_ values of 2.83, 3.6, and 5.3 µM, after 48 h exposure, for DON, 3-DON, and 15-ADON, respectively on HepG-2 [22]. Finally, with the same model, for ZEA and its derivatives, the values obtained after 72 h exposure were around 100, 15.2, and 119 µM for ZEA, β-zearalenol, and α-zearalenol, respectively [3]. Thus, according to IC_50_ data, ZEA is less cytotoxic, but showed interaction with estrogen receptors and interference with embryonic development and testosterone production compared to other mycotoxins [12]. Concerning emerging mycotoxins, enniatins (ENNs) and beauvericin (BEA), which are structurally similar molecules and both ionophores, were recently studied to determine their toxicity. Juan-García et al. (2013) showed that HepG-2 cell viability was significantly impacted after 72 h, with a reduction of 25% in cell numbers after exposure to ENN A and B [22]. Moreover, studies have also shown that after exposure to ENNs (24 h–48 h), multiple effects can occur, including increased ROS production and damage to the lysosome [23], lipid peroxidation, depolarisation of the mitochondrial membrane or deregulation of the cell cycle (particularly in the G2/M phase), and DNA damage [22,24]. In addition, Kalayou et al. (2015) demonstrated that concentrations higher than 10 µM ENN B caused significant reduction in testosterone (~84%), cortisol (~48%), and progesterone (~40%) in the H295R cell line. The same effect was found in Leydig cells, with a significant decrease in estradiol (~53%) and testosterone (~23%) levels [25]. To date, the most studied mycotoxins are ENN B and B1, because they occur at the highest levels in food and feed samples. However, some studies reported higher toxicity of ENN A1 or ENN A [24]. However, more data are required to understand the toxic effects and toxicity mechanisms of ENNs and BEA on human digestive tract cells, detoxification cells and sexual systems. Thus, in this study, we used 2D and 3D in vitro models with the HepaRG cell line. We determined mycotoxin cytotoxicity for three regulated *Fusarium* mycotoxins, namely T-2, DON, ZEA, and five emerging mycotoxins, namely ENN A, ENN A1, ENN B, ENN B1 and BEA, and compared their effects on both undifferentiated and differentiated cells. We also explored the transcriptomic response of HepaRG spheroids exposed to different ENN B1 and B concentrations to pinpoint the main pathways impacted by acute mycotoxin exposure.

## 2. Results

### 2.1. Cytotoxicity of Regulated Mycotoxins

For the cytotoxicity tests that were carried out on the HepaRG cell line, the tested mycotoxin concentrations were calculated, taking into account their solubility limits in solution. Mycotoxin exposure was performed for 48 h on two HepaRG models, either in monolayers (2D) or on spheroids (3D). For each model, undifferentiated and differentiated cell states were tested and compared. Cell viability significantly decreased in the presence of T-2, DON and ZEA. In the differentiated 2D model, viability was reduced by 89%, 77% and 73%, while in the undifferentiated 2D model, the reduction was 89%, 81% and 85% after 48 h exposure to T-2, DON and ZEA, respectively. For the 3D differentiated and undifferentiated models, 98% to 100% cell mortality was observed for T-2, DON and ZEA. Impact on cell viability was observed at 0.05 μM for T-2 (2D model differentiated/undifferentiated, −20%/−15%; 3D model differentiated/undifferentiated, −52%/−43%), 4 μM for DON (2D model differentiated/undifferentiated, −44%/−62%; 3D model differentiated/undifferentiated, −51%/−40%), and 45 μM for ZEA (2D model differentiated/undifferentiated, −21%/−47%; 3D model differentiated/undifferentiated, −96%/−72%). Overall, ICs for both 2D and 3D models were highly similar, and no significant variations were observed between undifferentiated and differentiated cells for T-2 and DON (Figure 1a,b). The IC_50_ values obtained for T-2 toxin varied between 0.04 and 0.12 μM for the 3D and 2D models with differentiated cells, respectively, and between 0.06 and 0.14 μM for undifferentiated cells, respectively. IC_50_ was very low according to the considered model and state, confirming the strong hepatotoxicity of this mycotoxin (Figure 1a). For DON, IC_50_ values ranged from 3.86 to 4.5 μM for the differentiated cell models, and from 2.14 to 4.09 μM for the undifferentiated cell models. DON was also strongly hepatotoxic, as IC_50_ ranged from 2.14 to 4.09 μM (Figure 1b). ICs obtained for the different models and cell states for T-2 and DON were not significantly different.

Finally, as expected, the IC_50_ concentrations of ZEA significantly higher than those of T-2 and DON to induce cell mortality. Different responses were observed depending on the considered model, with an impact on cell viability at a concentration as low as 15 μM for the 3D model, while 45 μM was needed for the 2D model (Figure 1c). Concerning the obtained ICs values, we observed more significant differences between results, in particular, for the same exposure concentration between the 2D and 3D models on differentiated and undifferentiated cells. The IC_50_ values obtained on differentiated cells were 12.18 μM and 48.09 μM on 3D and 2D models, respectively. With the same 3D conformation model but with a differentiated cell state, the IC_50_ values were significantly different. However, this was not observed for the 2D model, suggesting that the 3D model is more sensitive than the latter. After evaluating the cytotoxicity of regulated mycotoxins with our experimental conditions, we used the obtained results to compare emerging mycotoxin cytotoxicities in this study.

### 2.2. Cytotoxicity of ENNs and BEA Mycotoxins

For the emerging mycotoxins, ENN B, B1, A, and A1 and BEA all showed a dose-dependent effect on cell viability after 48 h exposure (Figure 2).

Overall reductions were 90.82%, 90.4%, 94.05%, 94.16% and 92% for the 2D model, while 92.52%, 97.27%, 97.35%, 97.52%, and 98.50% were observed for the 3D model for ENN B, B1, A, A1, and BEA, respectively. For the 2D model, a significant decrease was reached at 20.9 µM for ENN B (−85.77%) (Figure 2a), 20.4 µM for ENN B1 (−84.84%) (Figure 2b), 19.5 µM for ENN A (−84.61%) (Figure 2c), 5 µM for ENN A1 (−87.90%) (Figure 2d), and 8.5 µM for BEA (−85.10%) (Figure 2e). For the 3D model, the dose-dependent effect was significant from 20.9 µM for ENN B (−76.63%) (Figure 2a), 10.2 µM for ENN B1 (−41.24%) (Figure 2b), 0.0097 µM for ENN A (−20.45%) (Figure 2c), 5 µM for ENN A1 (−81.34%), (Figure 2d) and 0.085 µM for BEA (−23.15%) (Figure 2e). There were no significant differences between the 2D and 3D models for ENN B. Concerning the other mycotoxins, we observed no significant effects at very low or high doses. However, at intermediate doses, namely between 0.0085 and 0.85 µM for BEA, 0.00976 to 9.76 µM for ENN A, 0.025 to 0.25 µM for ENN A1, and 0.0102 and 1.02 µM for ENN B1, we observed a different sensitivity between 2D and 3D models in the middle of the range concentrations (Figure 2). Table 1 summarizes the IC_10_, IC_30_, and IC_50_ of BEA and ENN A, A1, B, and B1. Whether using the 2D or 3D model, significant differences were observed for the emerging mycotoxins for IC_30_ and IC_50_, which could be grouped into two clusters. The first cluster grouped ENN B (2D model: IC_50_ = 11.96 ± 2.20 µM; 3D model: IC_50_ = 15.17 ± 1.03 µM), ENN B1 (2D model: IC_50_ = 8.61 ± 4.66 µM; 3D model: IC_50_ = 14.58 ± 0.11 µM), and ENN A (2D model: IC_50_ =10.93 ± 4.80 µM; 3D model: IC_50_ = 13.89 ± 0.76 µM), which exhibited lower cytotoxicity, while the second cluster corresponded to ENN A1 (2D model: IC_50_ = 3.36 ± 0.21 µM; 3D model: IC_50_ = 3.86 ± 0.36 µM) and BEA (2D model: IC_50_ = 4.54 ± 1.73 µM; 3D model: IC_50_ = 6.71 ± 1.54 µM), with higher cytotoxic effects. Overall, the obtained ICs for the 2D and 3D models were not significantly different for emerging mycotoxins. The 2D model was reliable in terms of doses that could cause toxic effects, but only reflected toxic effects at high doses. The 3D model, closer to the real conformation of the target organ, was more sensitive to the low and medium concentrations tested, which are close to those found in contaminated foods. The 3D model thus seemed most appropriate for evaluating the post-exposure mycotoxin responses and obtaining the most representative ICs values.

### 2.3. RNA-Seq Analysis

As the 3D model can be considered to better represent in vivo reality compared to the 2D model, HepaRG spheroids were chosen to study the impact of enniatin exposure on gene expression. In this study, enniatins B1 and B were selected based on their prevalence in foods. To decipher the HepaRG cellular response to ENN B1 or ENN B acute exposure, RNA-seq sequencing was performed on spheroids (both differentiated and undifferentiated cells). To do so, we compared how gene expression was impacted for spheroids exposed to three concentrations of ENN B1 (at the IC_0_, i.e., sublethal concentration, IC_10_ and IC_30_ concentration) or one concentration of ENN B (IC_30_). First, quality assessment and alignment statistics for the gene expression data of controls, ENN B1, or ENN B treatments were verified, and results are shown in Appendix A. All samples obtained a mean read length of 149–150 bp and high-quality reads above 98.4%. Then, for each condition, we created a volcano plot to identify genes that were significantly over- or under-expressed (according to adjusted *p*-values) in the tested conditions in comparison to the control condition (after 48 h mycotoxin exposure) and according to the magnitude of change (log2 foldchange), which quantitatively described transcriptome changes. Comparisons were made two by two, with the corresponding cellular state (i.e., differentiated vs. undifferentiated cells). On Appendix A, volcano plot results showed the most up-regulated genes to the right, the most down-regulated genes to the left, and the most statistically significant genes on the upper section in red. Firstly, no differentially expressed genes (DEGs) were observed for the sublethal ENN B1 dose (IC_0_), regardless of the differentiated or undifferentiated cell state used. At 1.37 µM (IC_10_) ENN B1, different exposure responses were observed between the two cell states. For differentiated cells, a total of 241 DEGs were observed, with 142 genes down-regulated and 99 up-regulated, while for the undifferentiated state, only one gene (*MYH4*) was significantly over-expressed. When comparing ENN B1 and ENN B exposure at IC_30_, different genes were significantly and differentially expressed. Overall, a total of 489 and 484 DEGs were significantly expressed with differentiated cells in comparison to 124 and 311 for the undifferentiated cell state after ENN B1 and ENN B exposure, respectively. As shown on the Venn diagram (Figure 3), the nature of the up- and down-regulated genes varied according to the cell state.

Based on all results, we selected differentiated HepaRG spheroids as the best model to mimic real physiological cells in an in vivo state. Figure 4a represents the repartition of under-expressed genes, and Figure 4b over-expressed genes. No common genes were significantly and differentially over-expressed between the ENN B1/IC_10_, ENN B1/IC_30_, and ENN B/IC_30_ samples (Figure 4b). Nevertheless, we observed specific genes differentially for each sample; there were 85 over-expressed genes for ENN B1/IC_10_, 240 for ENN B1/IC_30_, and 247 for ENN B/IC_30_. In general, these genes belonged to the signal transduction, cell cycle, transcription and immune response pathways. Overall, 32 specific genes were significantly under-regulated for ENN B1/IC_10_, 116 for ENN B1/IC_30_, and 251 for ENN B/IC_30_ (Figure 4a). Forty-three genes were common to the three samples, with 18 being common to both tested ENN B1 concentrations (IC_10_ and IC_30_). When comparing ENN B and ENN B1 gene expression data at IC_30_, 54 genes were common. Based on all gene expression data for the three samples, the under-expressed genes were involved in the same pathways. We identified common pathways by analyzing gene expression between ENN B1/IC_10_, ENN B1/IC_30_, and ENN B/IC_30_. After comparing the expressed genes for each considered sample, an enrichment analysis was carried out using Enrichr to evaluate the most impacted pathways. For ENN B1/IC_10_, ENN B1/IC_30_ and ENN B/IC_30_ (Figure 4c–e), the main identified pathway corresponded to the complement and coagulation cascade, with respective *p*-values of 1.6 × 10^−7^, 4.8 × 10^−14^ and 1.4 × 10^−17^. This pathway is involved in the detection of soluble molecules sensing danger and triggering active enzymatic reaction cascades to provide a cell response [26]. Between the three samples, we also identified some common and significant pathways with a *p*-value < 0.05, such as the steroid hormone biosynthesis, the bile secretion pathway, retinol metabolism, and drug metabolism. Interestingly, the same transcriptomic response was observed for ENN B1/IC_10_, ENN B1/IC_30_, and ENN B/IC_30_, namely a significant down regulation of the genes implicated in the cholesterol biosynthesis pathway (*p*-values of 5.3 × 10^−3^, 2.6 × 10^−3^ and 5.3 × 10^−7^, respectively).

In order to further study the genes involved in the identified pathways, the under-expressed genes of each sample were obtained with Enrichr-KG software and are represented in Figure 5a. In general, the same genes were identified in the different samples, but an increase in the number of genes was observed between the ENN B1/IC_10_ and ENN B1/IC_30_. A common DEGs set was observed and corresponded to *UGT2B15*, *UGT2B10*, *UGT2B4*, and *UGT2B7*. The latter genes, linked to the microsomal membrane, are involved in conjugation by glucuronidation, using UDP-glucuronosyltransferases to catalyze various endogenous and xenobiotic compounds. Other genes common to the three samples belonged to the cytochromes P450 class. Modulated expression concerned the *CYP7A1* gene, coding for an enzyme involved in cholesterol metabolism and bile acid synthesis [27,28]; the CYP450 2C8 enzyme (*CYP2C8*), involved in xenobiotic metabolism, accounting for approximately 7% of total hepatic content [29]; and the *CYP8B1* gene, a monooxygenase involved in the absorption, distribution, metabolization and excretion (ADME) pathway, and the synthesis of cholesterol, steroids and other lipids [30]. *CYP2C19*, *CYP3A7*, *CYP2D6*, *CYP2A6*, *CYP2A9,* and *CYP1A2* were also found. Genes associated with the steroid hormone biosynthetic pathway were also impacted and belonged to the aldo-keto reductase (AKR) superfamily, containing the subclass of hydroxysteroid dehydrogenases (HSD), which are NADP(H)-dependent oxidoreductases. They are involved in steroid metabolism and also act as binding proteins or transmembrane bile acid transporters [31,32]. The bile acid pathway was also targeted. The bile acid and salt biosynthesis pathways are composed of 34 genes. In our study, we identified changes in gene expression for the *CYP7A1* and *CYP8B1* genes that are responsible for cholic acid synthesis. The genes *AKRC1C4*, *AKR1D1*, *ACOX2,* and *ABCB11,* which are implicated in metabolization of fatty acids and bile salt export, were also down-regulated. In view of the number of genes involved in cholesterol metabolism, we generated a heatmap (Figure 5b) for all the DEGs and compared them to the available database on cholesterol biosynthesis in GSEA software. Doing so, we clearly observed that the effects were less visible on undifferentiated cells versus differentiated cells. Expression of these genes in the differentiated spheroids exposed to ENN B1 at IC_10_ and IC_30_ was the most negatively impacted, followed by that of spheroids exposed to ENN B IC_30_. Out of a total of 27 genes involved in the cholesterol biosynthesis pathway, 23 genes, namely *DHCR24*, *MSM01*, *FDPS*, *DHCR7*, *HMGCS1*, *SREBF2*, *IDI1*, *FDFT1*, *SC5D*, *SQLE*, *HMGCR*, *LSS*, *NSDHL*, *ACAT2*, *MVD*, *SREBF1*, *LBR*, *PMVK*, *PLPP6*, *TM7SF2*, *ARV1*, *MVK*, and *HSD17B7*, were down-regulated, which implies that ENN exposure strongly impacted this pathway.

### 2.4. Quantification of Cholesterol and Cholesterol Esters in Spheroids

In order to confirm the results obtained from the RNA-seq analyses and, in particular, the notable down-regulation of the genes involved in the cholesterol biosynthesis pathway, we determined the total cholesterol content in spheroids after 48 h of exposure to IC_0_, IC_10_, IC_30_, and IC_50_ concentrations for both ENN B1 and ENN B, but also for three other mycotoxins of interest, namely ENN A, ENN A1, and BEA. The lowest impact on cholesterol was observed for ENN B, as only a 6.64% reduction was observed after IC_50_ treatment in comparison to the control sample. However, for all other mycotoxins, a significant difference in the amount of total cholesterol compared to the control was observed, even at the lowest IC_0_ concentration used, without a dose-dependent effect. On average, among the tested concentrations, we observed cholesterol reductions of 37.32%, 38.84%, 39.12%, and 40% for ENN A1, B1, A, and BEA, respectively, compared to the control (Figure 6). These results confirmed that ENN exposure impacts cholesterol levels in HepaRG spheroids.

## 3. Discussion

Nowadays, emerging mycotoxins are attracting greater interest due to their occurrence in the food and feed chain, and their potential threat to human health. Their toxicity, alone or in combination with other mycotoxins, as well as their mechanisms of action, cellular responses to exposure, and potential consequences on human health are still understudied. This is also the case for understanding their acute (short-term) and chronic (long-term) toxicities. The first aim of this study was to compare the toxic effects of emerging mycotoxins and compare them to regulated ones. Secondly, we also studied the impact of enniatin exposure on the transcriptome of HepaRG cells using spheroids, a more realistic in vitro 3D model, compared to classical 2D models. Overall, the obtained results showed that there was a mycotoxin-dependent cytotoxic effect. As ENNs and BEA have very similar structures, with only some R-group substitutions (more or less complex according to the molecule) on the main cyclic structure, this could induce a notable impact on their toxicity. Olleik et al. (2019) already highlighted the impact of these substitutions—with R1, R2, and R3 groups replaced by *iso*-propyl, *sec*-butyl, and phenylmethyl—on the toxicity of ENNs and BEA, suggesting that this aspect should be further studied [33].

Based on MTT assays, the cytotoxicity results showed that like the tested regulated mycotoxins (T-2, DON, and ZEA), emerging mycotoxins (ENNs and BEA) have a toxic effect on HepaRG cells in a dose-dependent manner (Figure 1 and Table 1). Similar ICs results have already been reported for regulated mycotoxins. As the obtained values show, the T-2 toxin had the most toxic effect. Similar values were found for the T-2-toxin in the study by Taroncher et al. (2020), with an IC_50_ of 0.061 µM (MTT assay). Fernández-Blanco et al. (2018) and Ivanova et al. (2006) reported an IC_50_ equal to 0.034 µM (MTT assay), and between 0.003 and 0.013 µM (AlamarBlue assay) and 0.008 and 0.009 µM (BrdU assay) after 48 h of exposure on HepG-2 cells, respectively [34,35,36]. Ivanova et al. (2006) and Fernández-Blanco et al. (2018) found similar toxicity for DON on HepG-2 liver cells after 48 h of exposure, the IC_50_ values being 1.4 µM (AlamarBlue assay), 3.5 µM (BrdU assay), and 2.82 µM (MTT assay) [35,36]. However, results can vary from one study to another; for example, on the same cell model after 48 h exposure, Cetin and Bullerman (2005) observed higher values, with an IC_50_ of 28.2 µM (MTT assay), while Zhou et al. (2017) found lower values, with an IC_50_ value of 0.23 µM (AlamarBlue assay) [37,38]. ZEA induces cell death only at high concentrations, but the main effects observed were strong estrogen and anabolic effects. The obtained values were consistent with various studies such as those of Smith et al. (2017), with an IC_50_ of 55.12 µM (MTS assay) after 48 h exposure on HepaRG cells; of Wang et al. (2014), with an IC_50_ of 39.88 µM (MTT assay); or of Marin et al. (2019), with an IC_50_ of 37.9 µM (MTT assay) after 48 h on HepG-2 cells [3,21,39]. Overall, in the present study, for T-2, DON and ZEA, the differentiated cell state was more sensitive than the undifferentiated one for the 2D model.

Concerning emerging mycotoxins, Ivanova et al. (2006) compared the cytotoxicity of ENNs and BEA to DON. After 24 h of treatment, the IC_50_s values differed according to the method tested. With the AlamarBlue assay, T-2 (0.003 µM) and DON (1 µM) were more toxic than BEA (8.8 µM, ENN B (206.7 µM) and ENN A (7.3 µM). Using the bromodeoxyuridine (BrdU) test, ENN A (1.6 µM), B (0.9 µM) and BEA (1.4 µM) showed a significant response at the same level as DON (3.3 uM) [35]. These values are in line with the results obtained, and underline the attractiveness of studying ENNs and BEA.

Our results suggested that ENNs and BEA have cytotoxic effects at concentrations that could be detected in contaminated samples with acute exposure. For BEA, similar IC_50_ values were determined by Ivanova et al. (2006) (between 8.8 and 22.2 µM -AlamarBlue assay- and 1.4 and 4 µM -BrdU-) and by Olleik et al. (2019) (3.4 µM after 48 h of exposure on HepG-2) [33,35]. The study by Meca et al. (2011) showed the same trend for ENN A, ENN A1, and ENN B in the HepG-2 cell line after 48 h exposure, with IC_50_ values of 11.4 µM, 2.6 µM, and 8.5 µM (MTT assay), respectively [40]. The ICs values obtained by Olleik et al. (2019) using Hep-G2 were lower, with a IC_50_ of 3.0 µM, 5.6 µM, 3.4 µM, and 5.6 µM (Rezasurin assay) after 48 h exposure to ENN A, A1, B and B1, respectively [33]. Results were more contrasting in the study by Ivanova et al. (2006) for ENN A, A1, B, and B1 after 48 h exposure on HepG-2, with the following IC_50_ values: 7.3–9.4 µM (AlamarBlue assay) and 1.6–2.5 µM (BrdU assay) for ENN A; 11.7–18 µM (AlamarBlue assay) and 2.6–4.2 µM (BrdU assay) for ENN A1; 206.7–435.9 µM (AlamarBlue assay) and 0.9–1.1 µM (BrdU assay) for ENN B; and lastly, 9.2–36 µM (AlamarBlue assay) and 2.8–3.5 µM (BrdU assay) for ENN B1 [35].

According to results from the literature, HepaRG cells treated with ENNs have similar sensitivity to intestinal cells (Caco-2), while CHO-K1 (ovarian cell), Jurkat-1 (T lymphocyte cell), SH-SY5Y (neuronal cell), and MRC-5 (embryonic fibroblast cell) cell lines are more sensitive. Moreover, in general, ENN A1 and B1 showed the highest toxic effect. The MTS assay used in this study is more sensitive and accurate than the MTT assay due to the formation of the formazan product without taking into account the cell debris that may be in suspension. All the results found in the literature are based on 2D monolayer models. No toxicity studies, either for regulated or emerging mycotoxins, have been carried out on more complex models such as spheroids in order to better represent the real structure of the organ. However, the 3D model is still expanding, and recent studies have focused on pollutants (i.e., flame retardants) and drugs (i.e., *Lumacaftor*) [41,42]. The 3D model has several advantages, such as a tissue-specific architecture that more closely resembles an in vivo state (in terms of morphology, polarity, cell communication, cell proliferation, and gene and protein expression) [43]. In addition, limited studies have been carried out to compare differentiated and undifferentiated cell states in in vitro models. Indeed, to make the best choice of in vitro model, both the structure and the state of cell development, which may impact cell sensitivity to mycotoxins, should be taken into account. In the tested conditions, we observed that the ENN and BEA emerging mycotoxins exhibited comparable cytotoxicity to that of DON, a regulated mycotoxin. Based on occurrence data of these mycotoxins, it is clearly of interest to continue research on these emerging mycotoxins using multiple mycotoxins simultaneously, as this situation best represents the reality of mycotoxin contamination.

To better understand the mechanisms of actions of these emerging mycotoxins and the cellular response to their exposure, we studied the transcriptome and the most-impacted signal pathways. Compared with other cell lines of human origin, in particular HepG-2, HepaRG cells are able to maintain liver-specific functions with the same level of expression found in primary human hepatocytes, such as cytochrome P450 enzymatic activities, or the formation of tight junctions involved in the formation of functional canalicular structures [44,45]. The HepaRG model is thus a very well adapted and relevant model for toxicological investigation. Moreover, in our study the cytotoxicity results showed that HepaRG 3D model is more relevant than the 2D model. Taking into account the results of over-expressed DEGs, the cell cycle pathway was significantly impacted by both enniatins. We identified CDK and CDC25C proteins (cell Division Cycle 25C), which have an important role in the cell cycle. However, we did not identify genes involved in the p53 pathway in response to DNA damage, as already described for another mycotoxin, aflatoxin B1 [46]. ENNs are able to disturb the normal progression of proliferating cells [26,47,48]. Various studies have highlighted the impact of ENNs on the cell cycle. ENN-treated HepG-2 cells showed a decrease in the S and G2/M phase and an increase in the SubG0/G1 phase, which is more marked in ENN A [22]. Prosperini et al. (2013) obtained similar results on Caco-2 cells (intestinal cell) with a G2/M phase arrest and a decrease in the G0/G1 phase but an increase in the SubG0/G1 phase [24]. Surprisingly, we obtained results showing an increase in DEGs involved in the cell cycle. Our results showed an increase in the expression of the *CCNB1*, *CDK1*, *CDC25C*, *CDC20*, *RRM2*, *GTSE1*, *CCNA2* genes involved in cell cycle arrest and DNA damage repair. Zhang et al. (2020) observed the same over-expressed genes belonging to the cell cycle, oocyte meiosis, and progesterone-mediated oocyte maturation pathways, and showed that some of these genes, notably *CCNB1* and *CDK1*, are related to progression in multiple tumor types [49]. Although carcinogenic effects have never been proven for ENNs in acute concentrations, it would be interesting to study in greater detail how cells regulate DNA repair after chronic exposure to ENNs and BEA. A larger amount of DEGs were down-regulated and common between the studied samples. The main impacted pathway was the complement cascade. It is involved in the rapid detection of toxic molecules in order to eliminate them. The results of this study suggested a decrease in genes involved in the formation of a membrane attack complex (MAC). MAC is composed of a complex of four proteins (C5b, C6, C7, and C8) and the accumulation of a fifth (C9) which forms a pore-inducing cell signaling response [50,51]. This pathway is involved in immune and inflammatory system responses. DEGs involved in steroid hormone pathways were also significantly down-regulated. In comparison with the steroid biosynthesis pathways described in the review by Schiffer et al. (2019), in the present study, all steroid pathways were impacted, and the following genes, *CYP11A1*, *CYP17A1*, *HSD11B2*, *SRD5A*, *SULT2A1*, *CYP19A1*, *HSD17B2*, and *AKR1C*, were down-regulated [52]. These down-regulated genes had an indirect effect on the steroid hormone biosynthesis pathway, impacting overall hormone production. Chiminelli et al. (2022) studied the effect of ENN A on ovarian bovine cell function, and concluded that steroid production was inhibited [53]. Other studies on fumonisin 1 and BEA in bovine cells showed a decrease in the expression of *CYP11A1* and *CYP19A1* genes [54,55]. Kalayou et al. (2015) showed that ENN B decreased the cell viability of Leydig and H295R cells, but also induced a significant reduction in the production of progesterone, testosterone, cortisol and estradiol [25]. Interestingly, in the same study, the level of transcribed genes also showed down-regulation of the *HMGR*, *CYP11A*, *HSD3B2*, and *CYP17A1* genes. In line with our results for cholesterol quantification, Chiminelli et al. (2022) showed a sudden but non-dose-dependent reduction in the amount of progesterone or estradiol on the small follicle GC [53]. The study by Payne and Hales (2004) showed the link between the cholesterol pathway and steroid hormones [56].

To the best of our knowledge, the results of the present study are the first to demonstrate the effect of ENN exposure on the cholesterol biosynthesis pathway, cholesterol being a precursor molecule of vitamins, steroid hormones, or bile acid biosynthesis. Bile acids are cholesterol derivatives, and their synthesis is the predominant metabolic pathway of cholesterol hydroxylation. They help modulate cell signaling pathways, stimulating secretion of pro-inflammatory cytokines, tumor necrosis factor α (TNFα), and interleukin-1β (IL-1β), and facilitating intestinal absorption and transport of lipids, nutrients, and vitamins [57,58]. We confirmed the involvement and relevance of cholesterol by quantitatively measuring cholesterol in exposed spheroids in comparison to the control spheroids. Our data clearly showed that even at sub-lethal concentrations, the quantity of total available cholesterol was reduced for ENNs A, A1, B1, and BEA. Although ENN B had an impact on transcription, no significantly quantifiable differences for cholesterol were observed. We hypothesize that it is necessary to take into consideration (i) the involvement of the steric hindrance and size of the molecule (which is directly linked to the R-group substitutions) and (ii) the toxicity of the molecule (ENN B had the less cytotoxic effect, and it is also the simplest chemical structure). Further tests based on these points need to be carried out. Figure 7 highlights the impact of the tested ENNs on the genes involved in cholesterol biosynthesis. In addition to being a precursor to multiple pathways, cholesterol is required for all cell membranes and helps maintain plasma membrane fluidity. The cholesterol pathway is also linked to steroid hormone synthesis and is essential for human health [56]. ENNs exposure could lead to negative outcomes or deregulation of cellular functions potentially modifying cholesterol homeostasis maintenance. Regarding literature data, some studies have already shown the impact of ENNs on lipid peroxidation or lysosome damage [23,24]. Olleik et al. (2019) observed lipid selectivity according to the considered mycotoxin, with ENN A, A1, B, and B1 having a preference for cardiolipin, a lipid present in mitochondrial membranes [33]. Very few studies on hypocholesterization have been performed. Porter and Herman (2011) reviewed the potential effects of cholesterol deficiency, showing that ENN B and B1 could have a major impact on the fetus [59].

## 4. Conclusions

In the present study, we first investigated the hepatotoxicity of several mycotoxins, either regulated (as references) or emerging mycotoxins, on different in vitro HepaRG models (2D monolayer vs. 3D spheroids) and using different cell states (undifferentiated vs. differentiated). Then, we evaluated the impact of ENN B and B1 exposure on HepaRG spheroids with differentiated cells. The IC_50_ values determined for the emerging mycotoxins were similar to those of regulated mycotoxins; thus, they may be a risk to human health and require further studies. Regarding cytotoxicity results, in the tested conditions, ENN A1 was the most toxic, followed by BEA, ENN A, B1, and B; this is potentially linked to the chemical structure of mycotoxins. We also demonstrated the relevance of the most sensitive 3D model. Despite structural similarities, ENNs and BEA differ slightly at the R1, R2, and R3 positions, likely explaining their different cytotoxicities. It would therefore be interesting to study the action and effects of different radical chains on cells. As different responses have been demonstrated for the tested mycotoxins, these dissimilarities seem to induce different action mechanisms, which have not yet been described. As a result, cell death occurs, notably affecting the expression of genes involved in cholesterol biosynthesis, bile acid biosynthesis, and steroid hormone biosynthesis. While the obtained results are of interest in the context of acute exposure, chronic exposure studies are also needed to better understand and represent the actual exposure of humans to foods contaminated by enniatins. Their cytotoxicity and cellular response to their exposure were studied here individually, while the reality of exposure corresponds to combinations with other mycotoxins and sometimes other detrimental compounds. Future studies should therefore include these combinations to better quantify the associated risk to human health.

## 5. Materials and Methods

### 5.1. Chemicals and Reagents

Beauvericin (BEA) (CAS 26048-05-5, purity ≥ 97%, PubChem CID: 3007984); deoxynivalenol (DON) (CAS 51481-10-8, purity ≥ 98%, PubChem CID: 40024); enniatin A (ENN A) (CAS 2503-13-1, purity ≥ 95%, PubChem CID: 57339252); enniatin A1 (ENN A1) (CAS 4530-21-6, purity ≥ 95%, PubChem CID: 57339253); enniatin B (ENN B) (CAS 917-13-5, purity ≥ 95%, PubChem CID: 164754); enniatin B1 (ENN B1) (CAS 19914-20-6, purity ≥ 95%, PubChem CID: 11262300); T-2 toxin (CAS 21259-20-1, purity ≥ 98%, PubChem CID: 5284461); and zearalenone (ZEA) (CAS 17924-92-4, purity ≥ 99%, PubChem CID: 5281576), were obtained from Sigma-Aldrich (St. Quentin Fallavier, France). Standards were dissolved in dimethylsulfoxide (DMSO) (Sigma-Aldrich) to final concentrations of 2.55 mmol/L for BEA, 3.374 mmol/L for DON, 2.93 mmol/L for ENN A, 2.99 mmol/L for ENN A1, 3.13 mmol/L for ENN B, 3.06 mmol/L for ENN B1, 2.144 mmol/L for T-2, and 6.282 mmol/L for ZEA, and stock solutions were stored at −20 °C. To prepare concentration ranges, DON, T-2 and ZEA were diluted in DMSO, while ENNs and BEA were diluted in water (Appendix A) and stored at −20 °C until use. William’s E medium, fetal bovine serum (FBS) and trypsin EDTA were obtained from Dominique Dutscher (Bernolsheim, France). Hydrocortisone (50 μM) and insulin ITS were obtained from Fischer Scientific (Illkirch, France). L-Glutamine was obtained from VWR (Fontenay-sous-Bois, France), while penicillin and streptomycin were obtained from Grosseron (Coueron, France). An RNeasy Mini Kit and RNase-Free DNase set were obtained from Qiagen (Courtaboeuf, France). CellTiter 96 Non-Radioactive Cell Proliferation Assay, CellTiter-Glo 3D Cell Viability Assay, and Cholesterol/Cholesterol Ester-Glo Assay kits were obtained from Promega (Charbonnières-les-Bains, France).

### 5.2. Cell Culture Conditions

The HepaRG cell line was obtained from Biopredic international (Saint-Grégoire, France). Cells were cultivated in William’s E medium supplemented with 10% of fetal bovine serum (FBS), 1% of penicillin/streptomycin antibiotics, 1% of 200 mM L-glutamine, 0.1% of hydrocortisone (50 umol/L solution in PBS 1X) and 0.1% of insulin ITS. Cells were incubated at 37 °C, 5% of CO_2_, and 100% humidity. The culture medium was changed every two to three days. When the cells were at confluence, cells were washed with PBS 1X and trypsinized. In detail, approximately 3 mL of trypsin were added to the cell monolayer and incubated at 37 °C, 5% CO_2_ and 100% humidity for 5 min. Trypsin activity was stopped with the addition of the same amount of FBS. Cells were centrifuged at 300× *g* for 5 min, the supernatant was removed, and then cell pellets were resuspended in 1 mL William’s E medium.

### 5.3. HepaRG 2D and 3D Models

To induce differentiation, 1% DMSO was added to the medium when cells were at confluence for 14 days. For both the 2D and 3D models, whether differentiated or undifferentiated cells, they were harvested in the same way as for culture conditioning with a PBS 1X for washing, trypsinization, centrifugation, and resuspension of the pellet. Cells were counted using a Malassez cell. As previously described by Gunnes et al. (2013) [60], cells were seeded at $5\times 10^{4}$ cells/well in a 96-well plate (Falcon, Corning Incorporated, NY, USA), and l in a U-bottom 96-well plate (PerkinElmer, Villebon-sur-Yvette, France) for the 2D and 3D models, respectively. For the 3D model, cells were incubated for 4–5 days at 37 °C, 5% CO_2_, and 100% humidity to form spheroids.

### 5.4. MTS Cytotoxicity Assay on 2D Model

The 2D model cells were exposed to DON, ZEA, T-2, ENN A, A1, B, and B1 and BEA individually at several concentrations (Appendix A). In addition, control wells containing only cells, DMSO or water were included. Plates were incubated at 37 °C, 5% CO_2_, and 100% humidity for 48 h. After incubation, cell viability was determined using a CellTiter 96 AQueous Non-Radioactive Cell Proliferation Assay (Promega, Charbonnières-les-Bains, France), according to the manufacturer’s instructions. The reaction consists of a bioreduction by cells of the MTS (3-(4,5-dimethylthiazol-2-yl)-5-(3-carboxymethoxyphenyl)-2-(4-sulfophenyl)-2H-tetrazolium inner salt) and PMS (phenazine methosulfate) mixture by cells into a formazan product that is soluble in tissue culture medium. Absorbance was measured using a Multiskan FC (ThermoFisher, Madison, WI, USA) at 450 nm. The concentrations inducing 10%, 30%, and 50% mortality (IC_10_, IC_30_ and IC_50_, respectively) were determined and used in succeeding assays.

### 5.5. 3D Cell Viability Assay

The 3D model cell was exposed to BEA, DON, ENN A, ENN A1, ENN B, ENN B1, T-2 and ZEA at the same concentration as 2D tests, and to respect the equilibrium constant, 2 µL of the mycotoxin solutions were added per well. In addition, control wells containing only cells, DMSO, or water were included as described above. The plates were incubated at 37 °C, 5% CO_2_ and 100% humidity for 48 h. At the end of the incubation period, cell viability was determined using a CellTiter-Glo 3D Cell Viability Assay (Promega, Madison, WI, USA), according to the manufacturer’s instructions. The assay consisted of a measurement of the number of viable cells with the luminometer, Luminoskan Ascent (ThermoFischer), based on the quantification of ATP, which is a marker for the presence of metabolically active cells. The homogeneous reagent causes cell lysis, then luciferin reacts with ATP, producing a luminescent signal proportional to the amount of ATP. The concentrations inducing 10%, 30%, and 50% mortality (IC_10_, IC_30_, and IC_50_, respectively) were determined and used in succeeding assays.

### 5.6. Cholesterol Quantification in Spheroids

In order to measure the amount of cholesterol in cells, the Cholesterol/Cholesterol Ester-Glo Assay (Promega, Madison, WI, USA) was used, and the reagent was prepared according to the manufacturer’s instructions. Spheroids were then washed carefully with 100 µL of PBS. Then, 50 µL of cholesterol lysis solution was added and shaken briefly before being incubated for 30 min at 37 °C. After incubation, 50 µL of cholesterol detection reagent was added. The plate was manually shaken for 30–60 s, and then incubated at room temperature for 1 h. To calculate the concentration of cholesterol, a range of total cholesterol standard samples were used to generate a standard curve. To confirm no interaction between ENNs and BEA with cholesterol, a negative control was tested. The luminescence was read using the luminometer Luminoskan Ascent (ThermoFischer).

### 5.7. RNA Extraction from Spheroids

For transcriptomic analysis, 10 days old spheroids, for both undifferentiated and differentiated cells, were exposed to either to ENN B1 (at IC_0_, i.e., sublethal concentration; IC_10_, i.e., 90% cell viability; and IC_30_ concentration, i.e., 70% cell viability) and ENN B (solely IC_30_ concentration) for 48 h (Appendix A). Total RNA was extracted from a pool of 12 spheroids using the RNeasy Tissue kit (Qiagen, Courtaboeuf, France) with an on-column DNAse treatment. Briefly, spheroids were transferred into Eppendorf tubes, then centrifuged at low speed (500× *g*) for 5 min. The supernatant was carefully eliminated and spheroids were directly frozen at −80 °C until extraction according to manufacturer’s instructions. RNA integrity was visualized after migration in a 1% DEPC-agarose gel electrophoresis, and RNA concentrations were determined using a Nanodrop 1000 (Thermo Fisher Scientific, MA, USA).

### 5.8. RNA Sequencing (RNA-Seq) Analysis

The RNA-seq FastQ files were pre-treated by Eurofins Genomics Europe Sequencing GmbH (Jakob-Stadler-Platz 7, Constance, Germany). Their quality was determined using Fastp software [61], and low-quality reads were removed and/or trimmed within the same tool. The reads were filtered to remove poor-quality bases within sliding windows having an average Phred score lower than 20. The adaptors within the reads were spotted and removed, and only reads longer than 30 bp were retained for the following analysis. The filtered reads within the FastQ files were aligned to the reference genome GRCh38 using the STAR (Spliced Transcripts Alignment to a Reference [62], by Sentieon [63] framework, while transcript quantification was performed by the featureCounts tool [64].

#### 5.8.1. Differential Gene Expression

The raw counts were then analyzed by differential gene expression (DGE) using the R package edgeR [65]. Statistical tests were performed for each gene to compare distributions between conditions (treatment vs. control), generating *p*-values for each gene. The threshold to consider the differentially expressed genes were set to Log2FC > |0.5| and FDR < 0.05. Nominal *p*-values were adjusted for multiple testing using the false discovery rate (FDR) using the Benjamin–Hochberg method. Using Ensembl genome browser, gene symbols were converted with hg38 reference genome.

#### 5.8.2. Pathway Enrichment Analysis

Gene ontology (GO) functional annotation and an enrichment analysis of KEGG pathways were performed using the Enrichr database [61,66]. Four categories, including Kegg 2021 for human pathways; biological process; molecular function; and cellular component, were included in the GO functional annotation. The 10 most-enriched GO elements and KEGG pathways were studied. The heatmaps were generated comparing the count database with the GSEA software channel database [67,68].

#### 5.8.3. Visualization of Differential Expression Analysis

To perform visual graphics, GraphBio, a new framework in R software (version 4.0.3, http://www.graphbio1.com/en/ (accessed on 13 May 2023) [69], was used.

### 5.9. Statistical Analyses

Raw data from cytotoxicity assessments of three independent experiments were collected and expressed as means ± SD. For cytotoxicity analysis, Tukey’s statistical test (* (*p* < 0.033), ** (*p* < 0.021), *** (*p* < 0.0002), **** (*p* < 0.0001), (ns = non-significant)) was used. For cholesterol concentration analysis, Dunnett’s statistical test was used (* (*p* < 0.033), ** (*p* < 0.021), *** (*p* < 0.0002), **** (*p* < 0.0001), (ns = non-significant)). Both were performed using GraphPad Prism version 9.5.1 (GraphPad Software, San Diego, CA, USA). The concentration inhibiting 50% of cell viability (IC_50_) was calculated from theoretical dose–response curves established with IC_50_ Tool Kit software using the GNUPLOT (Copyright 1986–1993, 1998, 2004, 2007 Thomas Williams, Colin Kelley) package. IC values were calculated from the following general Equation (1):(1)$$y=a+\frac{\left(b-a\right)}{\left(1+{\left(\frac{x}{c}\right)}^{d}\right)}$$
where *y* is the percentage of cell viability, “*a*” is the minimal percentage of cell viability, “*b*” is the maximum percentage of cell viability, “*c*” is the value of IC_50_, and “*d*” is the Hill coefficient.

Then, the latter formula was used to determine the concentrations inducing 10% and 30% mortality (IC_10_ and IC_30_, respectively). A non-cytotoxic concentration (i.e., the highest concentration of mycotoxins without cell mortality, named IC_0_) was also calculated for each mycotoxin, and used for RNA sequencing and cholesterol quantification analyses.

## Figures and Tables

**Figure 1 toxins-16-00054-f001:**
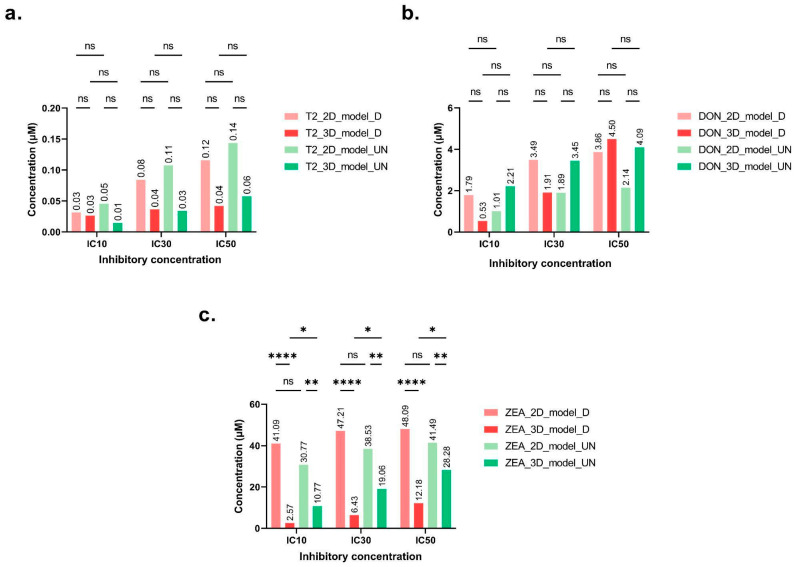
Inhibitory concentrations of T2-toxin (**a**), deoxynivalenol (**b**), and zearalenone (**c**) (µmol/M) (mean percentage ± SD of IC) after 48 h exposure. IC_10_, 3IC_30_, and IC_50_ values were calculated based on 10%, 30%, and 50% HepaRG cell death, respectively. HepaRG were cultured in monolayers (2D) or spheroids (3D) and using differentiated (_D) or undifferentiated (_UN) cells. After Tukey’s test, significant differences with the 100% viability control are indicated as follows: * (*p* < 0.033), ** (*p* < 0.021), **** (*p* < 0.0001) and (ns = non-significant).

**Figure 2 toxins-16-00054-f002:**
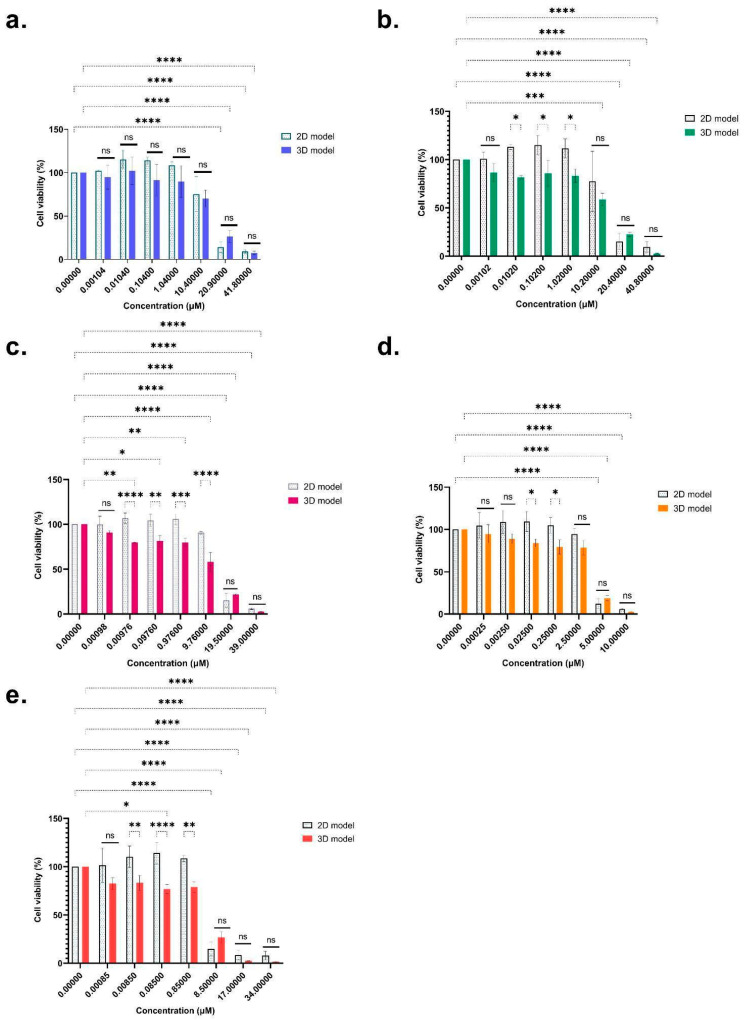
Cytotoxicity of ENN B (**a**), ENN B1 (**b**), ENN A (**c**), ENN A1 (**d**), and BEA (**e**) on differentiated HepaRG cells using monolayer (2D) and spheroids (3D), after 48 h of exposure (mean ± SD percentage of cell viability). After Tukey’s test, significant differences with the 100% viability control are indicated as follows: * (*p* < 0.033), ** (*p* < 0.021), *** (*p* < 0.0002), **** (*p* < 0.0001) and (ns = non-significant).

**Figure 3 toxins-16-00054-f003:**
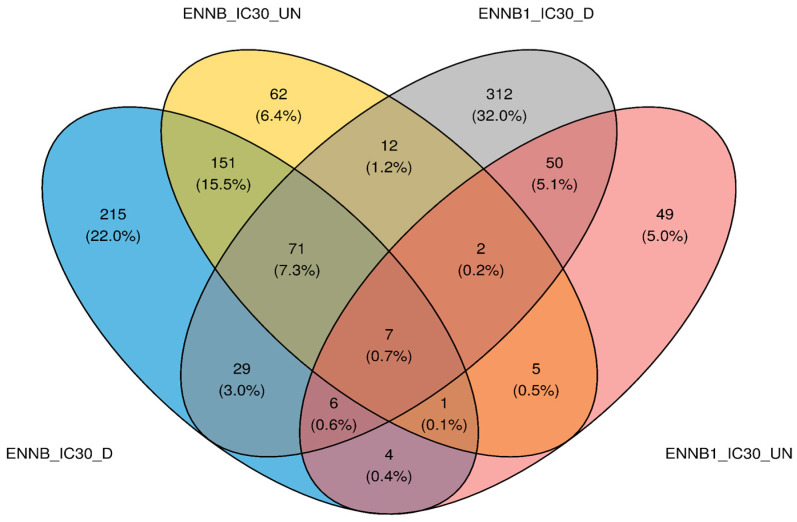
Number of genes differentially over- or under-expressed in comparison to the corresponding undifferentiated or differentiated negative controls. ENN B at IC_30_ is shown in blue, ENN B at IC_30_ is shown in yellow, ENN B1 at IC_30_ is shown in red, and ENN B1 at IC_30_ is shown in gray. HepaRG spheroids were cultured using differentiated (_D) or undifferentiated (_UN) cells.

**Figure 4 toxins-16-00054-f004:**
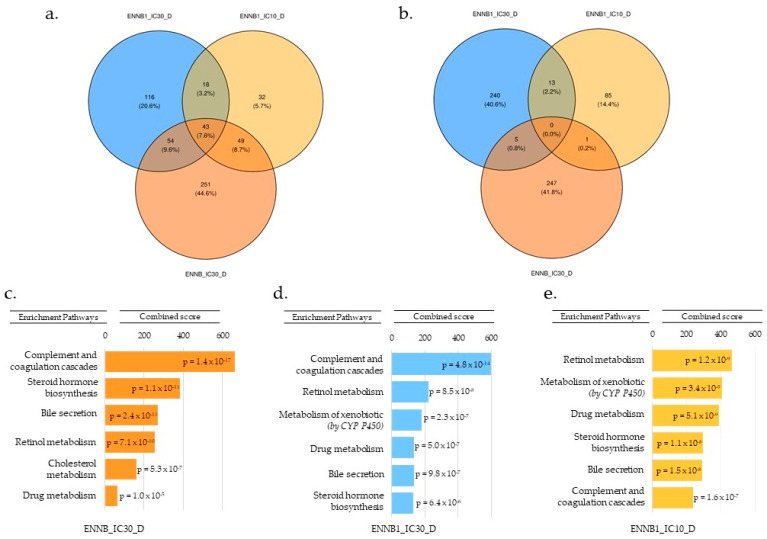
Number of genes differentially under- (**a**) or over-expressed (**b**) between ENN B/IC_30_, ENN B1/IC_30_, and ENN B1/IC_10_. Most significantly impacted pathways associated with down-regulated genes for ENN B/IC_30_ (**c**), ENN B1/IC_30_ (**d**), and ENN B1/IC10 (**e**), as determined by enrichment analysis using the Enrichr-KG software.

**Figure 5 toxins-16-00054-f005:**
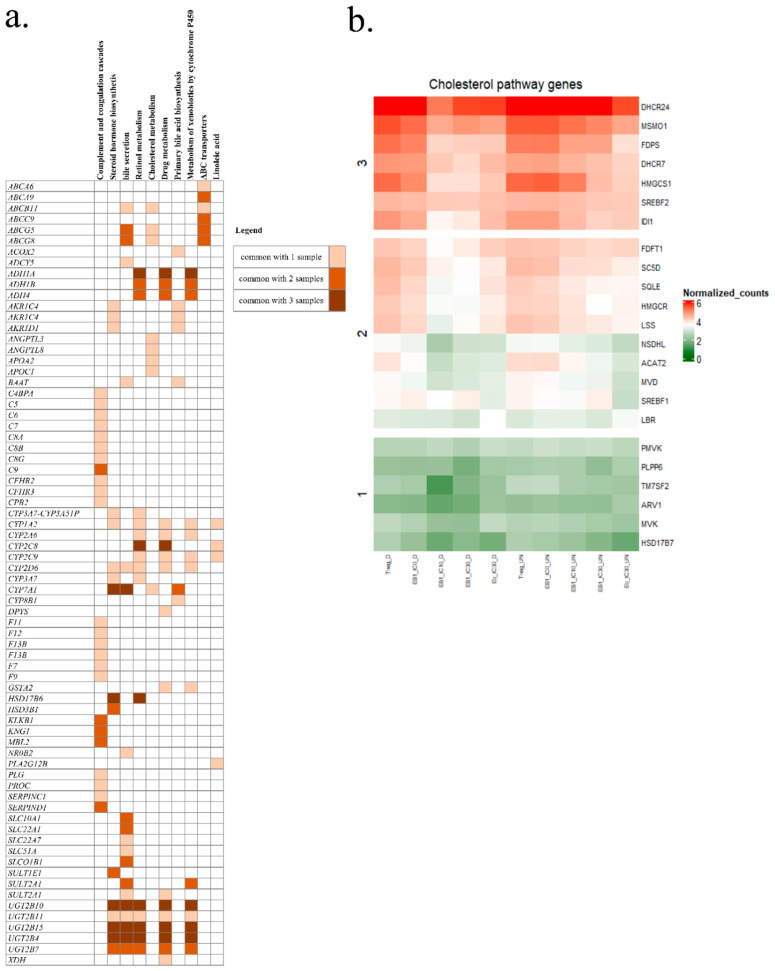
Significantly down-regulated genes belonging to different pathways of ENN B/IC_30_-, ENN B1/IC_30_-, and ENN B1/IC_10_-exposed samples (**a**), and a heatmap of the differentially expressed cholesterol pathway genes after mycotoxin exposure (**b**). The KEGGS 2021 library was used for the analysis.

**Figure 6 toxins-16-00054-f006:**
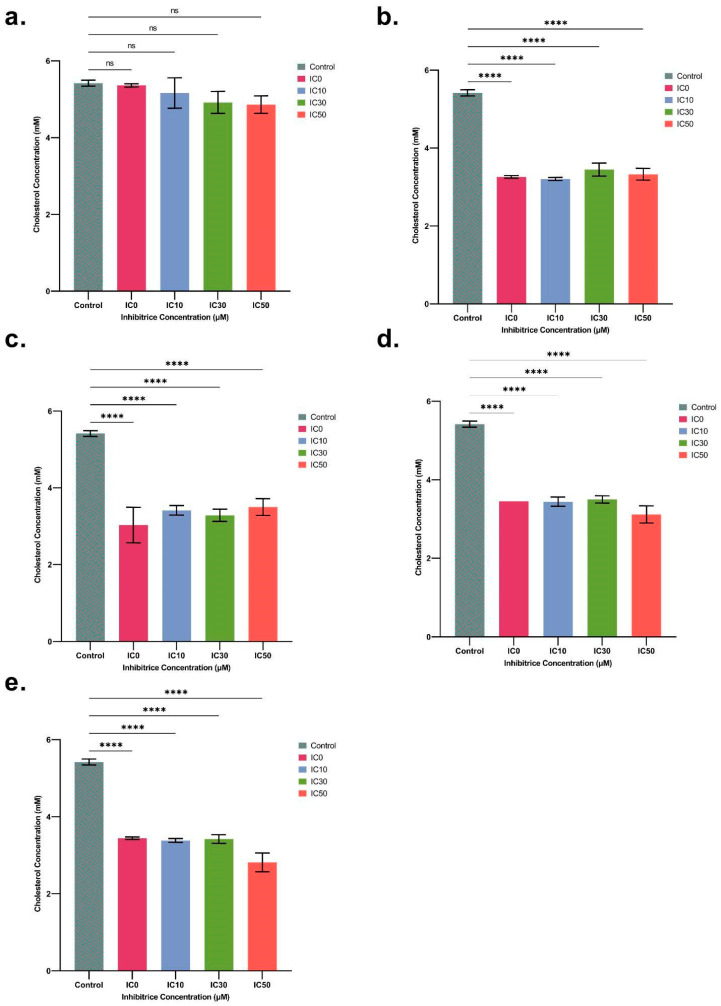
Cholesterol (mM) levels in spheroids after treatment with IC_0_ (a sub-lethal concentration), 10%, 30%, and 50% inhibitory concentrations (ICs) for ENN B (**a**), ENN B1 (**b**), ENN A (**c**), ENN A1 (**d**), and BEA (**e**) after 48 h of exposure (mean ± SD percentage of cell viability). After Dunnett’s test, significant differences from the control are indicated as follows: **** (*p* < 0.0001) and (ns = non-significant).

**Figure 7 toxins-16-00054-f007:**
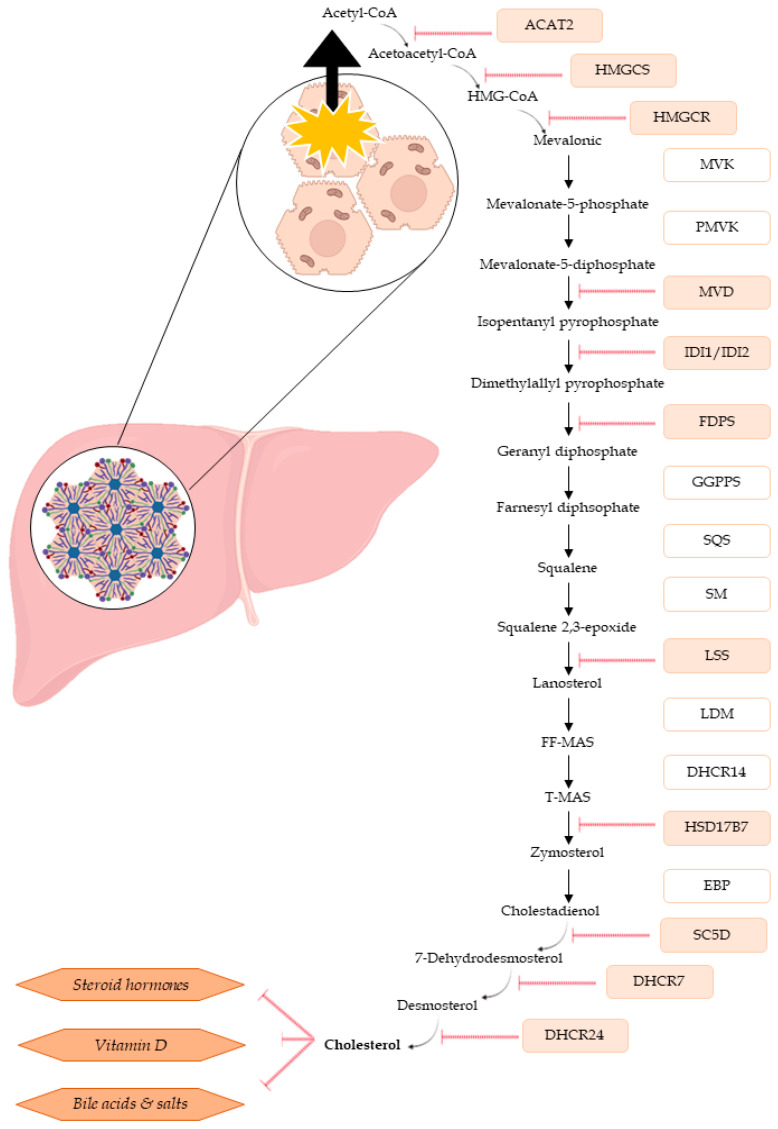
Diagram of cholesterol biosynthesis in hepatic cells and the different associated cellular pathways. All genes observed to be significantly down-regulated after ENN B and B1 exposure in this study are shown in red.

**Table 1 toxins-16-00054-t001:** Inhibitory concentrations of mycotoxins (µmol/M) (mean percentage ± SD of IC) calculated for 10% (IC_10_), 30% (IC_30_) and 50% (IC_50_) cell death levels of differentiated HepaRG cells cultured in monolayer (2D) and as spheroids (3D) after 48 h exposure time, (ns = non-significant).

Inhibitory Concentration	IC_10_ (μM)		IC_30_ (μM)		IC_50_ (μM)	
Model	2D	3D	2D	3D	2D	3D
T-2	ns		ns		ns	
	0.031	0.026	0.084	0.036	0.11	0.042
DON	ns		ns		ns	
	1.78	0.53	3.48	1.90	3.86	4.49
ZEA	ns		<0.0001		<0.0001	
	41.09	2.56	47.21	6.43	48.09	12.17
ENN B	ns		ns		ns	
	9.25 ± 2.02	7.25 ± 4.92	11.06 ± 2.15	10.39 ± 2.33	11.96 ± 2.20	15.17 ± 1.03
ENN B1	ns		ns		ns	
	6.19 ± 4.51	4.55 ± 2.38	7.83 ± 4.72	7.46 ± 1.74	8.61 ± 4.66	14.58 ± 0.108
ENN A	ns		ns		ns	
	7.88 ± 3.48	5.18 ± 2.75	9.76 ± 4.27	7.06 ± 2.43	10.93 ± 4.80	13.89 ± 0.76
ENN A1	ns		ns		ns	
	2.64 ± 0.18	2.29 ± 0.72	3.12 ± 0.19	2.98 ± 0.41	3.36 ± 0.21	3.86 ± 0.36
BEA	ns		ns		ns	
	3.30 ± 1.6	3.72 ± 2.56	4.14 ± 1.55	4.05 ± 1.65	4.54 ± 1.73	6.71 ± 1.54

## Data Availability

The authors confirm that the data supporting the findings of this study are available within the article [and/or its Appendix A].

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
