# Peer review of "Cytotoxic Effects of Major and Emerging Mycotoxins on HepaRG Cells and Transcriptomic Response after Exposure of Spheroids to Enniatins B and B1"

_toxins, 2024, doi:10.3390/toxins16010054_

Round 1

Reviewer 1 Report

Comments and Suggestions for Authors

The authors explored the cytotoxic effects of emerging fusariotoxins on HepaRG cells and measured the changes of genes upon the toxin exposure. The study demonstrated the effects of ENNs on cholesterol biosynthesis and the most impacted pathways. The findings provided new insights into the effcts of ENNs, which may contribute to the control of these mycotoxins. Some issues are concerned.

Line 251, three samples or three groups? It should be checked.

Lines 412-413, …the most impacted metabolic pathways? Or signal pathways?

Line 663, genes were set to log2FC>|0.5|, or |log2FC|?

Line 648, it should be checked.

How the statistical levels such as P<0.033, P<0.021 were defined?

In the discussion section, too many results were repeatedly described, it should be refined.

Author Response

Dear reviewer,

We greatly appreciate the time and efforts by the reviewer in reviewing this manuscript. We sincerely thank you for constructive criticisms and valuable comments which were of great help to revise this manuscript. We have answered all the points and carefully edited the paper and we feel that the quality of the paper has improved and hope you agree. All modifications in the text are shown in yellow.

Reviewer 1 :

Comments and Suggestions for Authors : The authors explored the cytotoxic effects of emerging fusariotoxins on HepaRG cells and measured the changes of genes upon the toxin exposure. The study demonstrated the effects of ENNs on cholesterol biosynthesis and the most impacted pathways. The findings provided new insights into the effects of ENNs, which may contribute to the control of these mycotoxins. Some issues are concerned.

Line 251, three samples or three groups? It should be checked.

Answer:

We are agree with reviewer. Transcriptomic analysis was performed on three samples.

(L250-252): No common genes were significantly and differentially over-expressed between the ENN B1/IC10, ENN B1/IC30 and ENN B/IC30 samples (Figure 4.b).

Lines 412-413, …the most impacted metabolic pathways? Or signal pathways?

Answer : 

(L 425-433) : “To better understand the mechanisms of actions of these emerging mycotoxins and the cellular response to their exposure, we studied the transcriptome and the most impacted signal pathways.”

Line 663, genes were set to log2FC>|0.5|, or |log2FC|?

Answer: 

Thank you for this comment.

(L 655): log2FC>|0.5|

Line 648, it should be checked.

Answer: 

(L 610-611): “The concentrations inducing 10%, 30% and 50% of mortality (IC10, IC30, and IC50 respectively) were determined and used in following assays.”

How the statistical levels such as P<0.033, P<0.021 were defined?

Answer: 

(L 672-678): “For cytotoxicity analysis, Tukey's statistical test (* (P < 0.033), ** (P < 0.021), *** (P < 0.0002), **** (P < 0.0001), (ns= non significant)) was used. For cholesterol concentration analysis, Dunnett's statistical test was used (* (P < 0.033), ** (P < 0.021), *** (P < 0.0002), **** (P < 0.0001), (ns= non significant)). Both were performed using GraphPad Prism version 9.5.1 (GraphPad Software, San Diego, California USA).”

In the discussion section, too many results were repeatedly described, it should be refined.

Answer: 

We have taken into consideration the redundancies. 

Reviewer 2 Report

Comments and Suggestions for Authors

GENERAL COMMENTS

The article addresses the contamination of cereals and cereal-based products by mycotoxins, focusing on emerging ones like enniatins (ENNs) and beauvericin (BEA). Emphasis is placed on Fusarium mycotoxins' high prevalence, causing toxic effects on the liver and intestine. Results reveal the cytotoxicity of T-2, DON, ZEA, ENNs, and BEA in HepaRG cells in both 2D and 3D models. Additionally, the transcriptomic study highlights disruptions in the cell cycle pathway and the complement cascade. The study underscores the significance of investigating emerging mycotoxins collectively to better reflect real-world contamination scenarios.

Analysis methods show adequacy, and results avoid undue interpretation. Yet, there's some unclear information requiring widespread verification. Manuscript publication could benefit from a minor revision.

SPECIFIC COMMENTS FOR REVISION

  • I have identified several typographical errors in the document, including issues with spaces, missing commas, dots, and variations in font styles and sizes. Some sentences are lengthy and complex; consider dividing them for clarity and ease of understanding.
  • Consistency in units of measurement throughout the article would be helpful; for instance, "ppb" and "µM" are used interchangeably. Maintain coherence in unit usage.
  • The figures require improvement; words appear blurry, making them difficult to read, and significance signals are barely visible. Particularly, Figure 4, the network, is challenging to interpret. Address these issues accordingly.
  • In the discussion section:
    • In some instances, results are presented without a clear connection to their biological significance. It would be useful to more thoroughly explain the biological relevance of certain findings and how they relate to human health.
    • The potential limitations of the study are not sufficiently discussed. It's crucial to address limitations for a better understanding of the scope of results and interpretations.
    • Despite the thoroughness of the discussion, it could benefit from greater synthesis to improve conciseness. Summarizing key results and main interpretations more succinctly is recommended, enhancing reader comprehension without compromising information quality.

Author Response

Dear reviewer,

We greatly appreciate the time and efforts by the reviewer in reviewing this manuscript. We sincerely thank you for constructive criticisms and valuable comments which were of great help to revise this manuscript. We have answered all the points and carefully edited the paper and we feel that the quality of the paper has improved and hope you agree. All modifications in the text are shown in yellow.

Reviewer 2:

Comments and Suggestions for Authors

GENERAL COMMENTS : The article addresses the contamination of cereals and cereal-based products by mycotoxins, focusing on emerging ones like enniatins (ENNs) and beauvericin (BEA). Emphasis is placed on Fusarium mycotoxins' high prevalence, causing toxic effects on the liver and intestine. Results reveal the cytotoxicity of T-2, DON, ZEA, ENNs, and BEA in HepaRG cells in both 2D and 3D models. Additionally, the transcriptomic study highlights disruptions in the cell cycle pathway and the complement cascade. The study underscores the significance of investigating emerging mycotoxins collectively to better reflect real-world contamination scenarios. Analysis methods show adequacy, and results avoid undue interpretation. Yet, there's some unclear information requiring widespread verification. Manuscript publication could benefit from a minor revision.

SPECIFIC COMMENTS FOR REVISION

I have identified several typographical errors in the document, including issues with spaces, missing commas, dots, and variations in font styles and sizes. Some sentences are lengthy and complex; consider dividing them for clarity and ease of understanding.

Answer: 

Thank you for your comments on typology problems. We apologize for the inconvenience. All the problems reported have been carefully corrected and verified.

Consistency in units of measurement throughout the article would be helpful; for instance, "ppb" and "µM" are used interchangeably. Maintain coherence in unit usage.

Answer:

We are agree with this comment.

(L 79-81): « Moreover, the average concentrations found were high with 548 µM for DON, 77 µM for ZEA, 98 µM for 3-DON, 192 µM for 15-ADON, 27 µM for ENN B1, 65 µM for ENN B, 14 µM for ENN A1, 4 µM for ENN A, 81 µM for HT-2 and 37 µM for T-2 toxin [18]. »

The figures require improvement; words appear blurry, making them difficult to read, and significance signals are barely visible. Particularly, Figure 4, the network, is challenging to interpret. Address these issues accordingly.

Answer: 

(L 315) Figure 5 has been replaced by a heatmap showing which genes are common to all three samples, and replaces the three networks for each sample to simplify understanding.

  • In the discussion section: 

In some instances, results are presented without a clear connection to their biological significance. It would be useful to more thoroughly explain the biological relevance of certain findings and how they relate to human health.

Answer: 

Thank you for this comment.

(L 495-498): “The cholesterol pathway is also linked to steroid hormone synthesis and is essential for human health [56]. ENNs exposure could lead to negative outcomes or deregulation of cellular functions potentially modifying cholesterol homeostasis maintenance.”

The potential limitations of the study are not sufficiently discussed. It's crucial to address limitations for a better understanding of the scope of results and interpretations.

Answer :

Thank you for this comment.

(L 518-520): “Regarding cytotoxicity results, in the tested conditions, ENN A1 was the most toxic, followed by BEA, ENN A, B1 and B potentially linked to the chemical structure of mycotoxins.”

(L 522-534): “It would therefore be interesting to study the action and effects of different radical chains on cells. As different responses have been demonstrated for the tested mycotoxins, these dissimilarities seem to induce different action mechanisms, which have not yet been described. As a result, cell death occurs, notably affecting the expression of genes involved in cholesterol biosynthesis, bile acid biosynthesis and steroid hormone biosynthesis. While the obtained results are of interest in the context of acute exposure, chronic exposure studies are also needed to better understand and represent the actual exposure of humans to foods contaminated by enniatins. Cytotoxicity and cellular response to their exposure was studied here individually, while the reality of exposure corresponds to combinations with other mycotoxins and sometimes other detrimental compounds. Future studies should therefore include these combinations to better quantify the associated risk for human health.”

Despite the thoroughness of the discussion, it could benefit from greater synthesis to improve conciseness. Summarizing key results and main interpretations more succinctly is recommended, enhancing reader comprehension without compromising information quality.

Answer: 

Thank you for this comment. We have taken into account the redundancies. 

Reviewer 3 Report

Comments and Suggestions for Authors

The current manuscript was well-written, but there are some issues that should be addressed before consideration for publication.

1. Both the 2D and 3D models, wether differentiated or undifferented cells were used in this study.  The authors should provide information to support they indeed successfully construted these cell models, especially 3D models. I did not found any reference in this manuscript that the research group have constructed thes cell models in their previous study.

2. Obviously, Fig S1 should be put in the manuscript but not in the supplementary. 

3. Indeed, it is not reasonable to study the toxicity of DON, ZEN and T2, as title of this manuscript is emerging mycotoxins. 

4. In figure 5, the error bar is not clear.

5.  For RNA-sequence analysis, was 2D or 3D cell model used. not clear. 

Author Response

Dear reviewer,

We greatly appreciate the time and efforts by reviewer in reviewing this manuscript. We sincerely thank you for constructive criticisms and valuable comments which were of great help to revise this manuscript. We have answered all the points and carefully edited the paper and we feel that the quality of the paper has improved and hope you agree. All modifications in the text are shown in yellow.

Reviewer 3 :

Comments and Suggestions for Authors

The current manuscript was well-written, but there are some issues that should be addressed before consideration for publication.

Both the 2D and 3D models, wether differentiated or undifferented cells were used in this study.  The authors should provide information to support they indeed successfully construted these cell models, especially 3D models. I did not found any reference in this manuscript that the research group have constructed thes cell models in their previous study.

Answer : 

We are agree with this comment.

(L 578-579) : « As previously described by Gunnes et al. (2013) [60], cells were seeded at 5.104cells/well in a 96-well plate (Falcon, Corning Incorporated, NY 14831 USA), and l in a U-bottom 96-well plate (PerkinElmer, Villebon-sur-Yvette, France) for the 2D and 3D models, respectively. »

Obviously, Fig S1 should be put in the manuscript but not in the supplementary. 

Answer : 

Thank you for this comment.

(L 157) Figure 1 is added.

Indeed, it is not reasonable to study the toxicity of DON, ZEN and T2, as title of this manuscript is emerging mycotoxins. 

Answer : 

(L 2-4) : « Cytotoxic effects of major and emmerging mycotoxins on HepaRG cells and transcriptomic response after exposure of spheroids to enniatins B and B1 »

In figure 5, the error bar is not clear.

Answer : 

Thank you for this comment.

(L 336) the error bars have been accentuated

For RNA-sequence analysis, was 2D or 3D cell model used. not clear. 

Answer : 

Thank you for this comment.

(L 213-216) : “As the 3D model can be considered to better represent in vivo reality compared to the 2D model, HepaRG spheroids were chosen to study the impact of enniatin exposure on gene expression. In this study, enniatins B1 and B were selected based on their prevalence in foods.” 

Round 2

Reviewer 3 Report

Comments and Suggestions for Authors

The manuscript has been improved and could be considered for publication.